# Peer review of "Exploring Molecular Mechanisms and Biomarkers in COPD: An Overview of Current Advancements and Perspectives"

_ijms, 2024, doi:10.3390/ijms25137347_

Round 1

Reviewer 1 Report

Comments and Suggestions for Authors

The authors by the title of this manuscript set out a huge task to review the insight obtained into the molecular pathways involved in the pathogenesis of COPD, addressing both initiation and progression of the disease.

However, the review reads too much like a review of reviews and lacks details on the molecular pathways involved. Sometimes, examples are given, such in the tables, but references are missing to original works, and one could have easily provided many other examples. 

Moreover, the pathogenesis itself is poorly described. Emphysema and chronic bronchitis are missing from the introduction, lung tissue remodeling is named, but it is not explained what this encompasses.

The importance of personalized medicine is emphasized, but the authors fail to stress the various aspects that are linked to the heterogeneity of the diseases that underlie this need.

Another aspect that could be stressed better is that when samples are used for multi-omic analyses into pathways, it is essential that a versatility of clinical characteristics are available to be able to link the data to personal traits in each patients.

Consider restructuring the manuscript as certain paragraphs are overlapping and iterative (eg epigenetics/genetics). Add schematics.

The challenges, future directions, and conclusions appear to take a somewhat different direction and appear more appropriate for a perspective vs a review. 

Comments on the Quality of English Language

I have no comments on the quality of the English

Reviewer 2 Report

Comments and Suggestions for Authors

The review by Li et al. summarizes aspects of molecular mechanisms that play roles in the development of COPD. The authors reference 87 articles from the field of COPD that well represent the recent knowledge in the field. The manuscript is well-written and touches on all major aspects of the subject. The conclusions are built on the basis of the discussion and emphasize the possible future direction of translational research on COPD.

Minor suggestion:

1)      Please consider listing the beta-2 adrenergic receptor as a molecular therapeutic target of COPD in Table 2 as a long-established, symptomatic, and effective therapeutical target (LABA).

Round 2

Reviewer 1 Report

Comments and Suggestions for Authors

The authors have taken the suggestions for improvement into consideration and the revised version of their manuscript now has a more clear story line.

However, I would still reconsider the title as it is suggested that the manuscript will provide a more detailed insight into the molecular mechanisms driving COPD than it really does. Along the same lines, it should be made clear that examples that are given are a selection and that many more biomarkers have been shown to be involved in COPD. This is true for for example oxidative stress markers, genetic markers through GWAS and blood based biomarkers. The examples provided in table 1 and 2 should be supported by references, so include a column with the appropriate reference for each marker and treatment.

Comments on the Quality of English Language

some minor editing is required
